# Rain-Based Train Washing: A Sustainable Approach to Reduce PM Concentrations in Underground Environments

Armando Cartenì [1], Furio Cascetta [2], Antonella Falanga [2],*  and Mariarosaria Picone [2]

[1]  Department of Architecture and Industrial Design, University of Campania "Luigi Vanvitelli",
     81031 Aversa, Italy; armando.carteni@unicampania.it
[2]  Department of Engineering, University of Campania "Luigi Vanvitelli", 81031 Aversa, Italy;
     furio.cascetta@unicampania.it (F.C.); mariarosaria.picone@unicampania.it (M.P.)
*    Correspondence: antonella.falanga@unicampania.it

**Abstract:** Fine particle concentrations measured in many underground rail systems around the world consistently exceed those observed at ground level, potentially posing significant implications for human health. While numerous authors have observed these high particle concentrations and analyzed both their atomic compositions and health impacts, few have investigated devices and technologies capable of reducing these high levels in underground environments. In light of these considerations and recognizing the multifaceted challenges associated with maintaining air quality in underground metro systems, the aim of this paper was to evaluate the usefulness and effectiveness of utilizing rainwater for washing trains to abate particulate matter (PM) concentrations in underground rail systems. To achieve this aim, an ad hoc case study was considered: the Naples Metro Line 1 (Italy), which is characterized by 4.5 km in the ground level and 13.5 km underground. A measurement campaign was carried out during storms of strong intensity through PM measuring instruments placed on station platforms along the metro line. Precisely, the trains were washed by the rain in the initial ground level section, and then continued wet within the underground one. The results of this measurement campaign were compared with those of a comparable survey carried out during average clear weather conditions, and the results showed that the train washing produces a significant $PM_{10}$ concentration reduction of up to about 60% in the underground environment. If confirmed in other experimental settings, these results could lay the groundwork for the introduction of structured washing system devices (e.g., periodically washing trains and/or tunnels) for the reduction of PM concentration in underground metro systems. The present study sought to contribute valuable insights towards sustainable and environmentally conscious approaches to addressing air quality concerns, particularly by harnessing the natural resource of rainwater during specific meteorological events.

**Keywords:** air quality; pollutant emission; particulate matter concentrations; underground metro system; washing effect; quality in public transport

## 1. Introduction

Within urban areas and cities, the transport system assumes a central role in the production of particulate matter (PM) emissions, responsible for 20–40% of the total pollutant and environmental emissions (e.g., [1–5]). To mitigate this negative incidence, urban planners emphasize the promotion of sustainable transport policies (e.g., [6–10]), such as the development of high-quality underground metro systems or the retrofitting of the traditional ones (e.g., [3,11–13]). Metro systems, which utilize an electrical power supply and move a high number of passengers/day, represent an effective solution to curbing greenhouse gas emissions. To enhance public transport utilization, factors such as service quality (e.g., environmentally friendly operations, passenger well-being, terminal aesthetics, assurance, and security) have emerged as pivotal elements in promoting urban sustainable mobility (e.g., [14–16]).

All around the world, numerous works of research consistently identified heightened particulate matter concentrations in subways. The increased PM concentrations primarily arise from the interaction between pantographs and catenaries. Furthermore, mechanical wear and tear, as well as friction during brake–wheel–rail contact, contribute to this high PM production. Finally, resuspension, induced by the piston effect and passenger movements within stations and trains, jointly with the deterioration of construction materials, further increase PM concentrations (e.g., [17–25]).

Despite the lack of specific decrees or legislation aimed at mitigating the emissions in subway system environments, many of the PM measurement surveys exceed the limits set at ground level by, for example, the European air quality directives for urban settings (i.e., 50 and 25 $\mu g/m^3$ for $PM_{10}$ and $PM_{2.5}$, respectively; [26,27]).

Although the available research results are not directly comparable due to differences in the subway system's age, ventilation system, tunnels, wheels category, braking mechanism, operating methods, measuring devices and techniques, and environmental factors (e.g., [3,11,25,28–31]), all the studies available in the literature, performed in several countries (Table 1), conclude that $PM_{10}$ concentrations at underground station platforms are systematically higher than those at the street level. For example, in Stockholm (2000) and Rome (2010), the average $PM_{10}$ concentrations on station platforms ranked among the highest worldwide, reaching average values equal to 469 $\mu g/m^3$ and 409 $\mu g/m^3$ (e.g., [32–34]) (with peak measures reaching 722 $\mu g/m^3$ and 877 $\mu g/m^3$, respectively). Other examples include Paris and Barcelona where $PM_{10}$ concentrations on station platforms were up to five times higher compared to those measured in the outdoor environments (e.g., [24,35,36]).

In additional case studies, the measured $PM_{10}$ concentrations were frequently found to be "exceedingly poor to moderately high", regarding the thresholds of the European air quality regulations for urban areas. The average measures range within 170 to 262 $\mu g/m^3$. For example, $PM_{10}$ concentrations of 288 $\mu g/m^3$, 262 $\mu g/m^3$, and 233 $\mu g/m^3$ were measured in Budapest, Beijing, and Stockholm, respectively [37–39]. Different seasonal concentrations, ranging between 166 and 275 $\mu g/m^3$, were observed in Tehran [35]. Comparable results were obtained in Milan and Naples (Italy), with $PM_{10}$ concentrations equal to 188 $\mu g/m^3$ and 195 $\mu g/m^3$, respectively [3,40]. Other examples with high measured $PM_{10}$ concentrations are found in Paris (220 $\mu g/m^3$), Prague (215 $\mu g/m^3$), Budapest (180 $\mu g/m^3$), Athens (172 $\mu g/m^3$), and Istanbul (170 $\mu g/m^3$) [41–45]. Finally, "medium–low" concentrations were measured in the underground metro systems of New York (142 $\mu g/m^3$—[46]), Seoul (108 $\mu g/m^3$—[47]), Athens (107 $\mu g/m^3$—[48]), Frankfurt (101 $\mu g/m^3$—[49]), Tehran (94 $\mu g/m^3$—[50]), Taipei (66 and 55 $\mu g/m^3$—[51] and [52], respectively), Los Angeles (78 $\mu g/m^3$—[53]), and Sydney (55 $\mu g/m^3$—[54]).

As has been said, high $PM_{10}$ concentrations in underground environments could produce serious problems for human health both for passengers and for train workers. Recent epidemiological studies have highlighted that the prolonged exposition to high $PM_{10}$ concentrations is closely linked to mortality imputable to ischemic heart disease, dysrhythmia, cardiac arrest, and heart failure (e.g., [55–62]). Research conducted by [63–67] investigated this issue, concluding that exposure to high $PM_{10}$ concentrations was also linked to pulmonary lesions and neurodegenerative anomalies. Ref. [68] provided evidence indicating that particles within subway environments exhibit a toxicity level around eight times higher, as also demonstrated by [69,70]. Furthermore, Refs. [71–73] concluded that high $PM_{10}$ concentrations could produce inflammatory damage to the human organism.

Hence, underground metro systems are highly unfavorable environments for humans. Evidence from the literature highlighted effective system devices to mitigate these high PM concentrations, including platform screen doors (e.g., [74,75]), powerful ventilation systems (e.g., [24]), and tunnel washing (e.g., [32,76]), as discussed in Section 2.

**Table 1.** Average $PM_{10}$ concentrations on subway station platforms in various underground rail metro systems (in parentheses is the year of the study).

| Case Study | Outdoor $PM_{10}$ [µg/m$^3$] | Station Platform $PM_{10}$ [µg/m$^3$] | References |
|---|---|---|---|
| Athens (2014–2012) | - | 107–172 | [44–48] |
| Barcelona (2011) | - | 346 | [24] |
| Beijing (2016) | 275 | 262 | [38] |
| Budapest (2014–2006) | - | 180–288 | [37–43] |
| Frankfurt (2013) | - | 101 | [49] |
| Istanbul (2007) | 70 | 170 | [45] |
| Los Angeles (2010) | 31 | 78 | [53] |
| Milan (2012) | 37 | 188 | [40] |
| Naples (2014) | 24 | 195 | [3] |
| New York (2021) | 29 | 142 | [46] |
| Paris (2013–2006) | - | 220–320 | [36–41] |
| Prague (2013) | - | 215 | [42] |
| Rome (2012–2010) | 32 | 407–409 | [22–34] |
| Seoul (2015–2004) | 43–155 | 108–359 | [32,47–77] |
| Shanghai (2013) | 190 | 457 | [33] |
| Stockholm (2017–2000) | 16–55 | 233–469 | [32–39] |
| Sydney (2015) | 20 | 55 | [54] |
| Taipei (2007–2016) | 60 | 55–66 | [51,52] |
| Tehran (2018–2011) | 72–139 | 94–275 | [35–50] |

Starting from these considerations, the aim of this paper was to investigate and quantify the usefulness of washing trains with rainwater (e.g., [78]) during storms of strong intensity (whose specific definition is reported in Section 2.3) in reducing $PM_{10}$ concentrations in underground metro systems. To achieve this aim, an ad hoc case study was considered: the Naples Metro Line 1 (Italy), which is characterized by 4.5 km in the ground level and 13.5 km underground. Precisely, a measurement campaign was carried out during storms of strong intensity through PM measuring instruments (portable photometric Aerocet 531 sample) placed on station platforms along the metro line. Precisely, the trains were washed by the rain in the initial ground level section, and then continued wet within the underground one. To assess the impact of this "washing effect", the results of this measurement campaign were compared with those of a comparable one carried out with average clear weather conditions.

## 2. Materials and Methods

### 2.1. System Devices to Reduce Particulate Matter Concentrations in Underground Metro Systems

Within the topic of sustainable mobility, although the literature abounds with case studies focusing on measuring elevated PM concentrations and their implications for human health, there is a noticeable lack of studies addressing effective devices able to reduce these concentrations in railway metro systems. This gap contrasts with the transportation planning literature, which emphasizes the importance of proposing strategic actions and projects to enhance the sustainable development of the transport system (e.g., [6–9]).

Within the topic of our research, only a few studies have investigated the usefulness of system devices in reducing PM concentrations in underground environments. For instance, as noted by [3,25,47], some "spatial factors" have been identified for retrofitting metro systems. One such factor is the implementation of platform screen door systems (Figure 1a) that could significantly reduce PM concentrations in station platforms, establishing a barrier between the platform and the train/tunnel, as observed in Taipei [51], Seoul [74], and Los Angeles [53]. This solution has been recently adopted in many metro systems around the world, both in new constructions and in the retrofitting of existing ones. For instance, until 2008, the Seoul underground railway system exhibited elevated $PM_{10}$ concentrations (359 µg/m$^3$—[77]). However, after implementing the platform screen door system, measurements by [74,75] showed a notable reduction in average $PM_{10}$ con-

centrations (103 and 97 $\mu g/m^3$, respectively). Analogous results were observed in the Barcelona subway, where a newly underground railway line equipped with platform screen doors measured particulate matter levels two/three times lower than those in a traditional metro system (e.g., [24]). Furthermore, there are other additional useful tools for reducing elevated concentrations of microparticles, as highlighted in [3]. Among these, there are the high-quality (powerful) ventilation systems (Figure 1b) and the washing of the station platforms, trains, and tunnels (Figure 1c,d), that allow us to reduce particle resuspension, which is one of the main causes of these high PM concentrations in the underground environment (e.g., [7,24,32,51,52,74–76]). For example, Ref. [32] investigated the tunnel "washing effect" in the Stockholm subway system. The authors, employing water to wash the tunnel surfaces and railway lines between the platforms of the subway system, observed that, during a few days, the concentrations were reduced by approximately 13%. This result suggests that new particles originating day by day from the tunnel surfaces and tracks play a relatively minor role in influencing the overall $PM_{10}$ concentrations, compared to the resuspension of particles caused by the passage of trains and generated in the previous weeks/months (and which can be removed from the tunnels with washing). Additionally, Ref. [76] investigated the impact of the tunnel washing on PM levels. The results revealed a reduction of about 46% in $PM_{10}$ concentrations 3.5 months after the tunnel washing. This post-cleaning $PM_{10}$ reduction could be linked to porous tunnel walls acting as particle sinks and/or the air-conditioning system filtering effect. Finally, by removing these particles from surfaces and preventing them from becoming airborne, the practice of washing trains/tunnels can contribute to cleaner and healthier air in the underground environments. Furthermore, when combined with other measures, such as an efficient ventilation system, it can be a comprehensive strategy for addressing air quality concerns in underground metro systems.

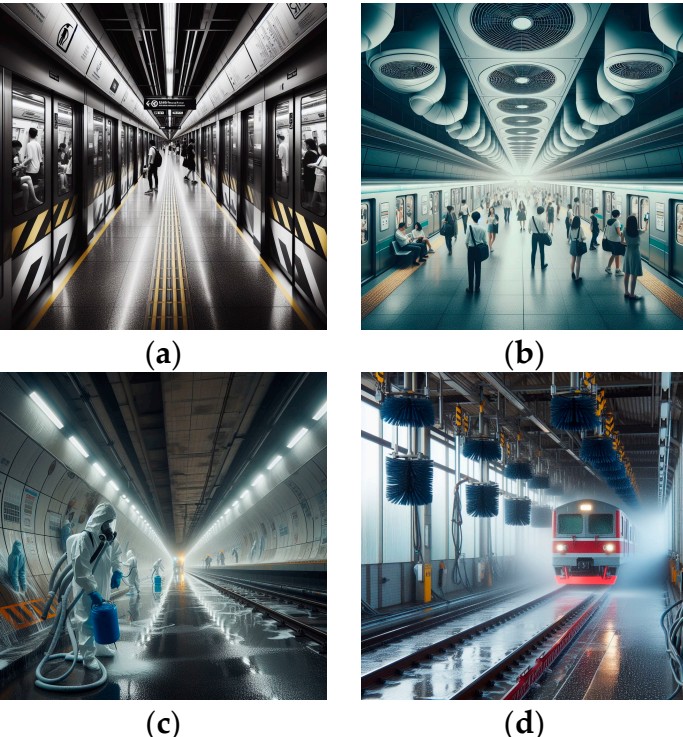

(**a**)　　　　　　　　(**b**)

(**c**)　　　　　　　　(**d**)

**Figure 1.** (**a**) Example of platform screen door system; (**b**) example of ventilation system; (**c**) example of tunnel washing; and (**d**) example of train washing (source: exemplary images generated through AI).

### 2.2. Case Study

As stated above, the proposed ad hoc case study is the Naples Metro Line 1 (Figure 2). Naples, situated in southern Italy, is a historic city renowned for its rich cultural heritage and stunning coastal views. With a population of about 1 million and a density of 8566 people/km², Naples is one of the largest cities in Italy, offering ideal conditions for a railway-centric public transportation system that aligns with Transit-Oriented Development principles (e.g., [79–81]). Firstly, the dense urban fabric and high population density of Naples underscore the importance of public transportation for mobility and road congestion reduction. Secondly, Naples' geographical features, characterized by its coastal location and surrounding hills, necessitate a transportation network capable of navigating varied terrain and connecting disparate neighborhoods.

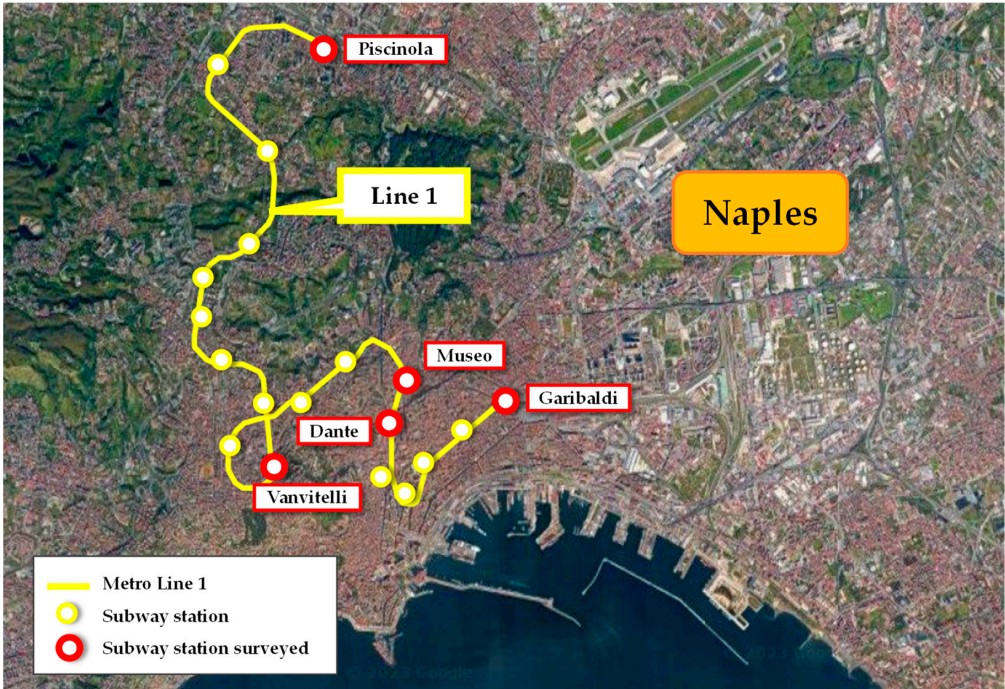

**Figure 2.** The Line 1 of the Naples metro system.

The Naples Metro Line 1, also known as the "Collinear Metro", stands as one of the primary arteries in the city's public transport system. It connects the northern part of the city (Piscinola/Scampia) with the main multimodal rail station Garibaldi, the major interchange transport node of the city, with both national and high-speed rail services. Weekdays see approximately 242 train trips/day from 6:00 to 23:00, with an average frequency of one train every 8 min, operating at an average speed of 32 km/h. It serves about 135 thousand travelers on weekdays and 50 thousand on weekends and holidays.

With a length of approximately 18.7 km, the Line 1 weaves its way through diverse areas of Naples, operating both in underground and at ground level landscapes. Notable for its path following the city's orographic profile, the line encounters varying altitudes and terrains. It presents 19 stations (3 in ground level and 16 underground). Beginning in the northern part of the city, named Piscinola, situated at a relatively higher elevation compared to the historic city center, Line 1 travels southward through residential and commercial districts while gradually descending along its path. En route, the line passes through different areas, including historic neighborhoods such as the Historic Center of Naples, distinguished by its narrow streets and ancient structures (Figure 2).

The line mainly operates underground, mitigating traffic congestion and respecting the densely constructed nature of the surroundings. Line 1 emerges above ground level in the peripheral neighborhoods, traversing open expanses and industrial zones. This elevated stretch provides passengers with a panoramic view of the city and its environs.

The journey ended at Garibaldi Station, a pivotal railway and transportation hub located near the historic center and major suburban rail services. Overall, Line 1 of the Naples Metro serves as a crucial link connecting diverse areas of the city, along varying elevations, and offering passengers a travel experience that mirrors the distinctive topography and geography of Naples.

Based on the representation proposed in [40], the features of the stations are outlined in Figure 3, detailing aspects such as tunnel type, platform depth, and surrounding urban traffic conditions.

| Station | Tunnel type | [m] | Platform depth [m] | Outdoor ambient |
|---------|-------------|-----|--------------------|-----------------|
| Piscinola | | 0 | +15 | Urban – low traffic congestion |
| Vanvitelli | | 8.305 | -18 | Urban – low traffic congestion |
| Museo | | 12.504 | -16 | Urban – high traffic congestion |
| Dante | | 13.010 | -25 | Urban – low traffic congestion |
| Garibaldi | | 18.009 | -40 | Urban – high traffic congestion |

Station platform

Ground-level railwat (outdoor)

Two-ways tunnel railway

One-way tunnel railway

**Figure 3.** Features of the subway systems surveyed of Naples Metro Line 1 and the area nearby the station entry.

Figures 4 and 5 illustrate the layouts of the metro stations involved in the survey, in addition to the configuration and operational mode of the ventilation and air-conditioning systems. This is relevant because the separation form between the public area (including platform and station hall) of the subway station and the tunnel of the section, in addition to the form and operation mode of the ventilation system, could affect the number of fine particles from different sources entering the public area of the station.

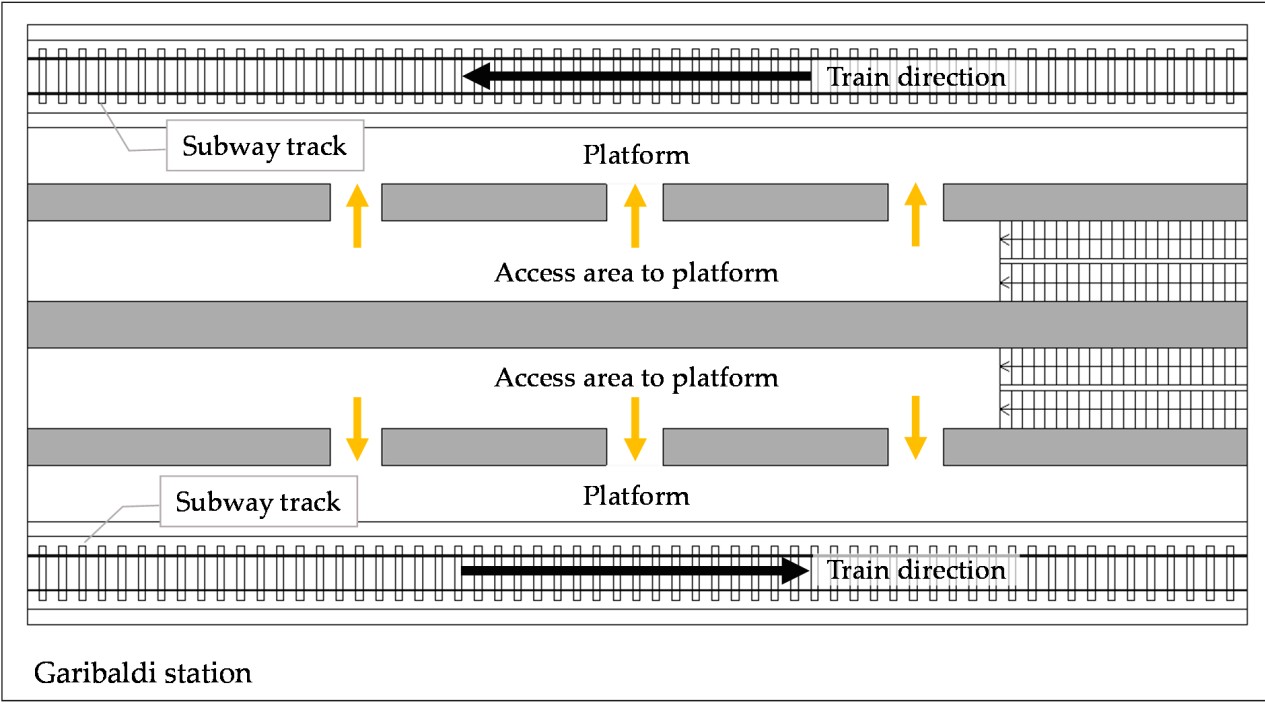

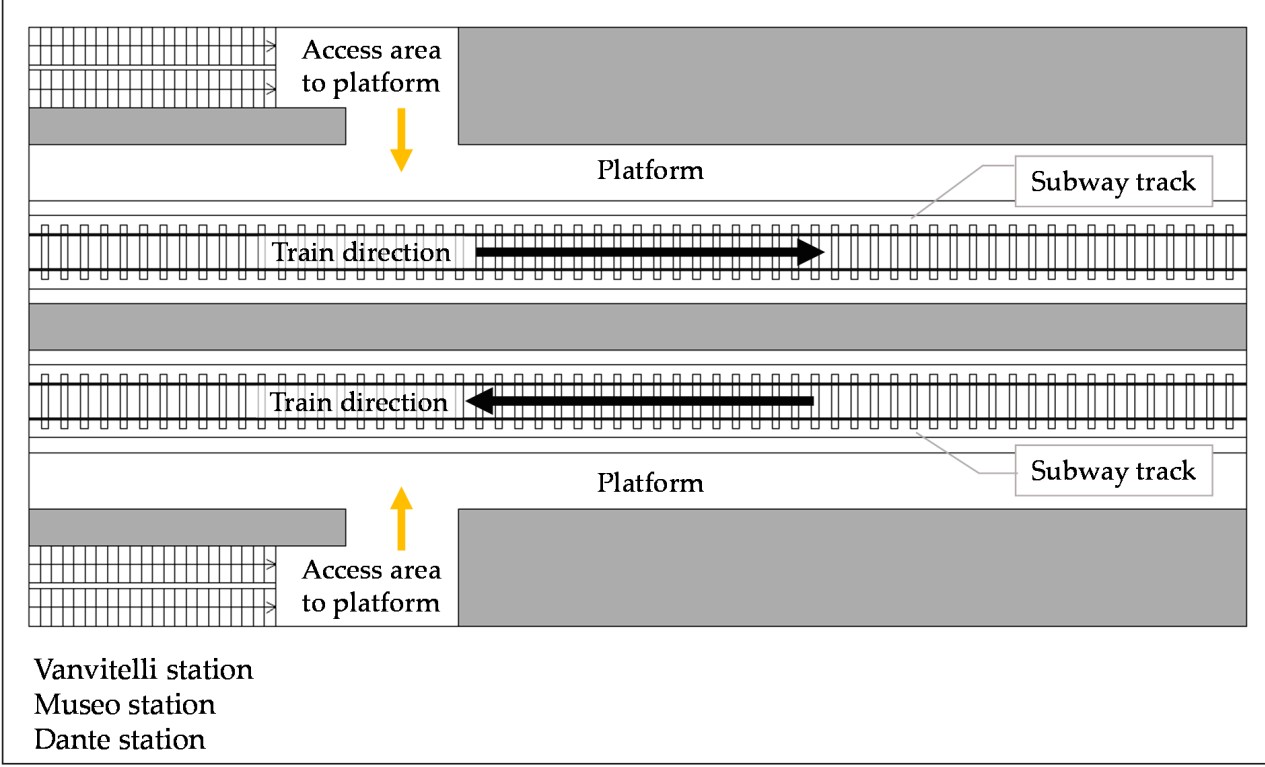

**Figure 4.** The layouts of the metro stations involved in the survey.

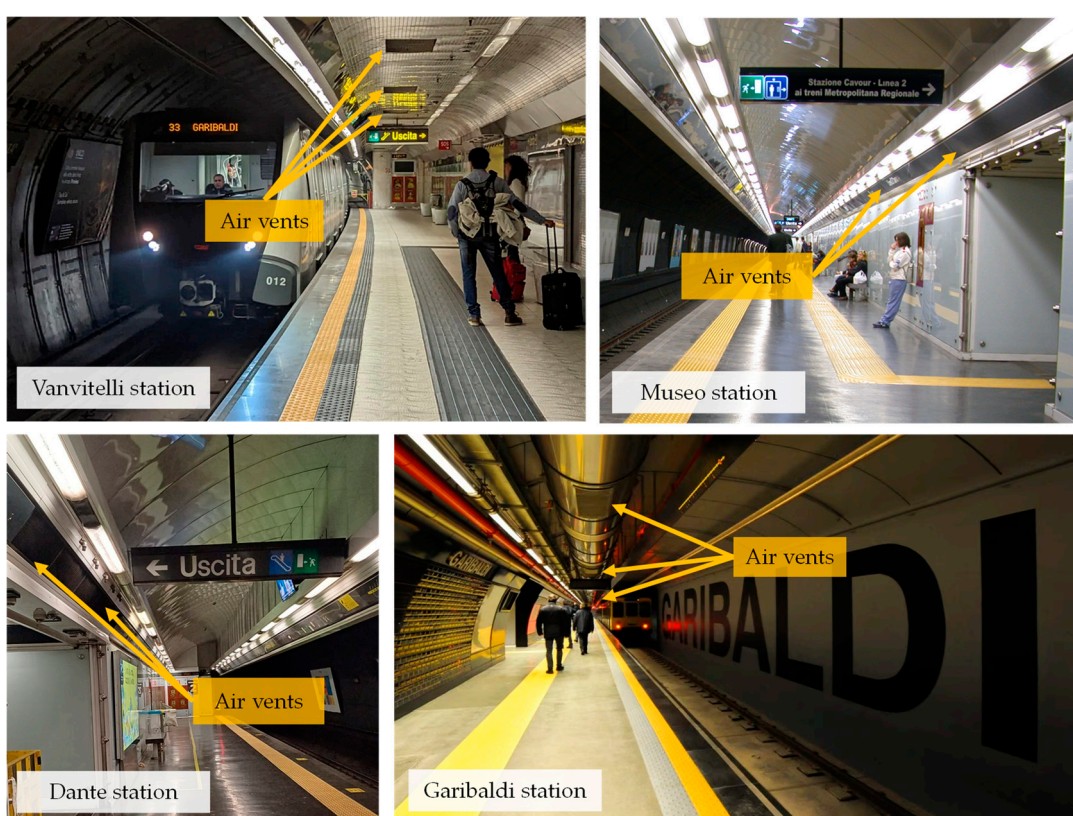

**Figure 5.** Configuration and operational mode of the ventilation and air-conditioning systems of the metro stations surveyed (source: [82]).

### 2.3. Monitoring Method

Taking into account the brief sampling duration (e.g., before and during the train arrivals) and the live measurements performed in different station platforms, the instrument used to measure $PM_{10}$ concentrations was the portable photometric Aerocet 531 sampler (for more information, refer to [83,84]). This device is an automatic device capable of measuring $PM_1$, $PM_{2.5}$, $PM_7$, $PM_{10}$, and total suspended particulates, using a time interval of 2 min.

Consistent with comparative literature studies, this measurement device is frequently used for assessing PM concentrations, both inside or outside (for instance, see references [3,85–91]). To minimize potential inaccuracies in measurement, the Aerocet 531 device underwent calibration utilizing certified data provided by the Campania Environmental Agency (CEA). The CEA unit is a LSPM10 airborne particulate continuous monitor and a sampler (UniTec, Auckland, New Zealand, 2014), and was factory-calibrated and furnished with a gravimetric module consisting of 16 membrane filters, each with a diameter of 47 mm. This configuration enables particulate sampling rigorously along the lines of EN12341/14 standard [92]. In most of the monitoring stations in the Campania region, these analyzers are frequently utilized and give hourly average data, which provides real-time information on pollution levels. With the aim to calibrate the photometric $PM_{10}$ concentrations measured with the Aerocet 531 with those measured by the CEA unit, four outdoor CEA units near the monitored stations and one indoor CEA unit were considered.

Mass conversion is made using standard conversion factors or using suitable factors based on unique conditions (K-Factor). K-factor was estimated regressing Aerocet 531 $PM_{10}$ measures on the corresponding gravimetric units. $PM_{10}$ data were reliable with a determination coefficient of about 90%. Calibration results allow us to adjust $PM_{10}$ values measured with the Aerocet 531 through the equations (K-factor):

$$\text{Corrected } PM_{10} = 1.16 \times \text{Measured } PM_{10}$$

Refs. [86,90] obtained calibration factors comparable to the estimated one.

Furthermore, ambient aerosol particles can change their physical and chemical properties, including size and refractive index, by either absorbing or releasing water in response to changes in relative humidity at the atmosphere. Consequently, in environments with elevated humidity levels, it becomes essential to comprehend the processes of water uptake and release by aerosol particles to address issues related to light scattering (e.g., [93]). Studies have noted that exceptionally high ambient relative humidity levels can induce unusually high readings of particulate matter when using certain sampling techniques (e.g., [94]). For instance, as demonstrated by [93,95], elevated relative humidity values lead to inaccuracies in measurements conducted by light-scattering particle analysis devices. Ref. [94] suggested the application of a specific correction factor to adjust particulate matter levels when relative humidity exceeds 70%. However, in our case study, because of the measured relative humidity was consistently below 55%, which indicates a dry environment, it was not necessary to take any specific measures to adjust the particulate matter readings.

### 2.4. Measurement Campaigns

As said, to investigate the usefulness of washing trains with rainwater during storms of strong intensity to reduce $PM_{10}$ concentrations in subways, two measurement campaigns were carried out at station platforms during a morning rush hour within the peak period between 7:00 a.m. and 9:30 a.m. Four business days for both the survey were selected in December 2022 and January 2023, characterized by different weather conditions, collecting both the "dry sample" (average seasonal clear weather conditions) and the "wet sample" (adverse weather conditions during storms of strong intensity). Furthermore, the logging interval for the surveys was set at 2 min, surveying 120 sample data (1 rush hour × 4 days × 30 samples/h) for each station and survey (wet and dry). The instrument was always located in the center of the platform.

Both the surveys were performed on the station platform of five main metro stations of Line 1 (one in ground level, Piscinola, and four in underground environments, Vanvitelli, Museo, Dante, and Garibaldi). More specifically:

1. the first campaign was performed on clear weather conditions with surface temperatures ranging within the seasonal average values (12–18 °C), low wind conditions (wind speed < 10 km/h), and outdoor relative humidity ranging from 50% to 60% (Table 2). During this survey, the average measured temperatures on the underground station platform were moderately higher (16–17 °C) than those at street level and the relative humidity was, as said, always lower than 55%;

2. the second campaign was performed during a storm of strong intensity, with surface temperatures ranging between 10–15 °C and mid-high wind conditions (wind speed > 10 km/h), with a relative humidity ranging from 65% to 85% (Table 2). Precisely, this survey was carried out for one hour within the morning peak period during a storm of strong intensity and starting the measures 30 min after the onset of adverse weather conditions.

To make the experiment repeatable in other contexts, it was defined as "storm of strong intensity", an atmospheric phenomenon accompanied by one or more of the following weather phenomena (source: [96]): point cumulation greater than 40 mm/1 h, intensity greater than 20 mm/15 min, large hailstorms (grain diameter > 1 cm), high number of lightning strikes, and violent blows of wind and/or whirlwinds.

Furthermore, some preliminary pre-tests were also performed with variable sampler height. Precisely, these pre-tests were performed in the middle station of the line "Museo"; the instrument was always located in the center of the platform and four different sampler heights were tested: 1.40 m, 1.50 m, 1.60 m, and 1.70 m. The logging interval for the surveys was set at 2 min, surveying 30 sample data during a rush hour for each sampler height tested. Results indicate no significant average differences between measures relative to the different height (not reported for reason of brevity).

**Table 2.** Outdoor environment conditions measured in dry and wet conditions (December 2022–January 2023) (source: [97]).

| Outdoor Environment Conditions | Dry Conditions | Wet Conditions |
|---|---|---|
| Surface temperature | 12–18 °C | 10–15 °C |
| Wind speed | 3–9 km/h | 14–26 km/h |
| Relative humidity | 50–70% | 65–85% |

## 3. Results and Discussion

The main results of the surveys are reported in Table 3 and in Figure 6. The average $PM_{10}$ concentration measured in dry conditions ranged from 20.7 to 191.6 μg/m$^3$ with a standard deviation from 6.2 to 33.5 μg/m$^3$. By contrast, the average $PM_{10}$ concentration measured in wet conditions is significantly higher, ranging between 16.0 and 99.4 μg/m$^3$ with a standard deviation of between 4.0 and 28.5 μg/m$^3$ (Table 3). To verify the statistical difference between the values measured in dry and wet conditions, a sample t-test was conducted with a confidence level set at 95%. The results show that concentrations measured in dry conditions are significantly different from those measured in wet conditions (results are not reported for reason of brevity).

**Table 3.** Distribution of indoor (station platform) $PM_{10}$ mass concentrations (μg/m$^3$) measured in dry and wet conditions (December 2022–January 2023).

| Station Platform | Dry Sample | | | | Wet Sample | | | | (Wet − Dry)/Dry |
|---|---|---|---|---|---|---|---|---|---|
| | Min | Average | Max | St. Dev. | Min | Average | Max | St. Dev. | Average Percentage Variation |
| Piscinola | 15.0 | 20.7 | 47.0 | 10.1 | 12.0 | 16.0 | 24.0 | 4.0 | −22.6% |
| Vanvitelli | 132.0 | 176.3 | 220.0 | 33.0 | 50.0 | 73.9 | 88.0 | 12.1 | −58.1% |
| Museo | 141.0 | 179.9 | 255.0 | 33.5 | 54.0 | 82.5 | 127.0 | 23.8 | −54.1% |
| Dante | 175.0 | 183.0 | 190.0 | 6.2 | 72.0 | 84.5 | 100.0 | 10.6 | −53.8% |
| Garibaldi | 172.0 | 191.6 | 220.0 | 17.6 | 58.0 | 99.4 | 145.0 | 28.5 | −48.1% |

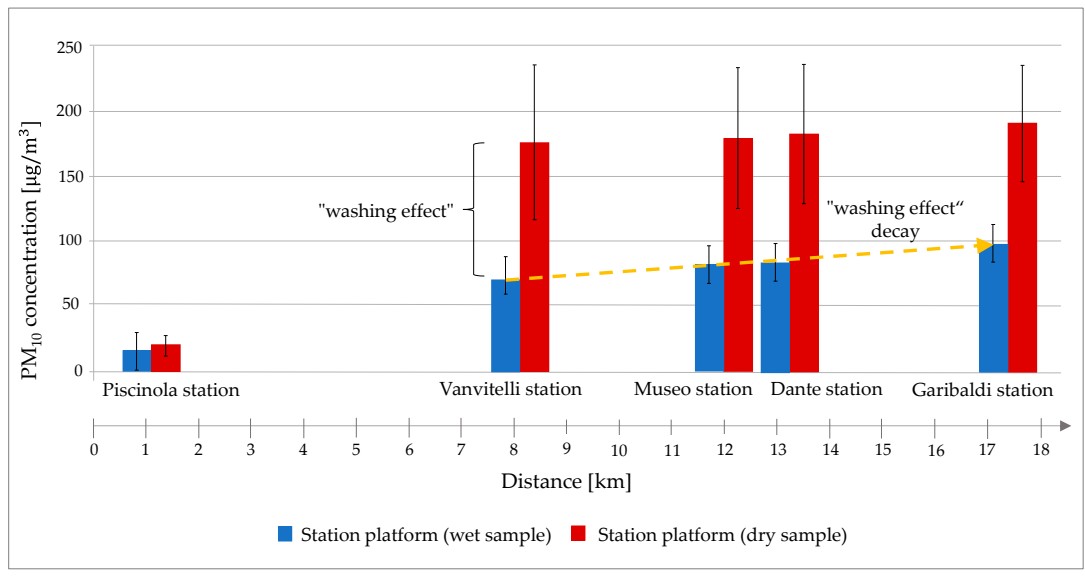

**Figure 6.** Distribution of indoor (station platform) $PM_{10}$ average mass concentrations (in μg/m$^3$) measured in wet and dry conditions (December 2022–January 2023).

These findings indicate a significant reduction between the concentrations measured during the wet conditions against the dry ones, with the percentage difference ranging

from $-22.6\%$ to $-58.1\%$. These results are comparable to those obtained, for example, by [76], who measured a 45.9% reduction in $PM_{10}$ concentrations after the washing of the metro system tunnel in Taiwan.

It is interesting to note how this observed "washing effect" gradually decreases as the train moves along the underground section (as highlighted by the yellow dotted arrow in Figure 6). For example, the $PM_{10}$ concentrations measured in wet conditions are about 58.1% lower than those measured in dry conditions at the first Vanvitelli station investigated (the closest to the ground level section); by contrast, this reduction decreases up to the 48.1% at the final Garibaldi station of the line.

## 4. Conclusions

Much of the research carried out around the world has extensively documented the presence of elevated PM concentrations within underground metro systems. This result has raised significant concerns within the scientific community due to the potential health risks associated with prolonged exposure to fine particles, which can adversely affect respiratory health and exacerbate conditions such as asthma and cardiovascular disease.

Despite the wealth of literature addressing this issue, there is a relevant gap in specific studies in terms of useful system devices able to reduce PM concentrations in underground metro systems. While these devices are discussed in the literature, there is a lack of comprehensive research elucidating their effectiveness in reducing PM concentrations and mitigating associated health risks. Considering this gap, the primary objective of this study was to investigate the efficacy of the trains' "washing effect" through rainwater during storms of strong intensity. This experimental field aimed to assess the potential benefits of utilizing natural resources to mitigate PM concentrations within underground metro systems, with a specific focus on Naples' Metro Line 1 (Italy).

To achieve the research aim, a comprehensive survey campaign was planned and executed in December 2022 and January 2023 in Naples. Through data collection and a comparative analysis between stormy and clear weather conditions, the study revealed a significant reduction in $PM_{10}$ concentrations by up to 60%, as a result of the train washing. This notable reduction in $PM_{10}$ concentrations underscores the potential effectiveness of utilizing rainwater to mitigate air pollution within underground metro networks. Moreover, the study found that the cleaning impact of rainwater washing was particularly pronounced when the train entered the first underground station ($-58\%$), gradually decreasing, with up to a 48% reduction, upon reaching the last station.

These promising findings are specific to the Neapolitan case study but, if confirmed in other comparable case studies (i.e., characterized by metro lines partially in the ground level and partially underground), could lay the basis for future public transportation construction and suggestions for relevant government departments. Among these, there could be, for example, the periodic train and tunnel washing to effectively mitigate the PM levels and/or the adoption of the platform screen door system to create a barrier between the high PM concentrations and the passengers waiting for the train.

Furthermore, such research holds significant implications for public health and technological innovation, as it directly impacts the well-being of passengers, nearby residents, and workers.

Starting from the findings of this research, some technological suggestions for dust reduction for the case study of Naples could also be individuated as, for example, the development of a platform screen doors system or a high-quality ventilation system connected to a PM monitoring system that amplifies filtration when PM levels exceeded attention limits.

Moreover, future research developments also aim to assess the influence of external wind speed at the ground level and atmospheric humidity conditions on the estimated "washing effect". It is expected that outdoor wind can only marginally influence PM concentration levels at station platforms, as these stations are very deep underground (between 16 and 40 m). By contrast, humidity probably could influence the results of the measurements since ambient aerosol particles can absorb or release water in response

to changes in relative humidity. This could alter their physical and chemical properties, affecting their light-scattering behavior.

Finally, future research efforts in this field should prioritize optimizing existing technologies, developing novel solutions, and evaluating the long-term impacts on health and the environment. In summary, investigating and implementing systems to reduce PM concentrations in subways offer tangible benefits for urban community health and well-being, while also advancing scientific and technological progress in air purification and environmental sustainability.

**Author Contributions:** Conceptualization, methodology, validation, formal analysis, supervision, project administration, writing—original draft preparation, and writing—review and editing, A.C.; conceptualization, methodology, validation, supervision, project administration, writing—original draft preparation, and writing—review and editing, F.C.; resources, formal analysis, data curation, writing—original draft preparation, and writing—review and editing, A.F.; resources, formal analysis, data curation, writing—original draft preparation, and writing—review and editing, M.P. All authors have read and agreed to the published version of the manuscript.

**Funding:** This research received no external funding.

**Institutional Review Board Statement:** Not applicable.

**Informed Consent Statement:** Not applicable.

**Data Availability Statement:** Data are contained within the article.

**Conflicts of Interest:** The authors declare no conflicts of interest.

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
