# Peer review of "Rain-Based Train Washing: A Sustainable Approach to Reduce PM Concentrations in Underground Environments"

_sustainability, doi:10.3390/su16072708_

Round 1
Reviewer 1 Report
Comments and Suggestions for Authors
Dear Author/Authors,
Please consider the following recommendations:
1. Introduction
Lines 62 – erase this (the topic investigated in this research) is unnecessary, already known;
Line 77 you have Tehran, line 82 is Teheran. Please use the same type of naming.
Line 118 – 122, The structure of the document unfolds as follows: Section 2 outlines materials and methods, encompassing the devices employed to mitigate elevated particulate matter concentrations in underground metro systems and detailing the experimental case study.
Section 3 delves into the outcomes of the measurement campaigns. Ultimately, Section 4 offers the conclusions reached.
must be erase. It is not necessary to present the sections of the work, they can be seen quickly.
2.2. Case study
Lines 201- 203: Garibaldi, Duomo, University, City Hall, Toledo, Dante, Museum, Materdei, Sal-201 vator Rosa, Four Dyas, Vanvitelli, Gold Medals, Montedonzelli, Rione Alto, Policlinico, 202 Aminei Hills, Frullone, Chiaiano, Piscinola.
This information is unnecessary, must be erase.
Lines 250-276: The Aerocet 531 aerosol particulate monitor (Figure 4) combines particulate mass monitoring and particle counting functionalities within a single device (further information can be found in [87]). This device is compact, practical, and furnished with a portable rechargeable battery (6V Ni-MH)……..
………………………………
including Figure 4. The Aerocet 531 aerosol particulate monitor instrument.
This information’s is unnecessary, must be erase (too much).
3. Discussion
You can insert some technological suggestions for dust reduction for the case analyzed in Naples.
Regards,

Reviewer 2 Report
Comments and Suggestions for Authors
Moderate changes are needed in English; the language could be improved slightly, and the style would benefit from a spell check.
Reviewer 3 Report
Comments and Suggestions for Authors
1.The authors said “A measurement campaign was carried out during adverse weather conditions; the trains were washed in the initial ground level section and then continued with their journey in the underground..”, It seemed that the adverse weather condition was only rain, which I did not think such weather will impose too much influence on the condition.
2. Table 1 required more explanations. For instance, the last two rows seemed from same authors. It was recommended to provide more information by re-organizing the table and corresponding contents.
3. Please try replace figure 1 with more fruitful images.
4 The authors were recommended to add a section which provide details about method used in the study.
5. The authors were required to provide more information about dry samples, wet samples in table 2 and figure 5. For instance, how many wet samples did you collect, where did you collect the samples, etc.
6 The following studies were recommended to be properly cited: [1] Rain to Rain: Learning Real Rain Removal Without Ground Truth, IEEE Access, vol. 9, pp. 57325-57337, 2021, doi: 10.1109/ACCESS.2021.3072687. [2] Ship imaging trajectory extraction via an aggregated you only look once (YOLO) model, Engineering Applications of Artificial Intelligence, vol. 130, 2024, doi: 10.1016/j.engappai.2023.107742.
Comments on the Quality of English LanguageEnglish should be improved.
Reviewer 4 Report
Comments and Suggestions for Authors
The theme and analysis of this paper are not in the scope of sustainability and the presented research does not present the required contribution to become a paper in a renowned scientific journal. To reach this level further work will become necessary, considering the analysis of the outdoor environment (wind temperature and humidity), larger periods of analysis, and many others (the definition of what exactly means the conditions of rainwater during thunderstorms and dry weather). In my view, this is an interesting preliminary work.
Comments on the Quality of English LanguageSome minor problems are easily corrected and Figure 1 does not add value to the paper, so it can be replaced by a figure explaining the way measurements were made (taking out the photo 4 - equipment).
Round 2
Reviewer 2 Report
Comments and Suggestions for Authors
The language description in the article can be more refined,and the sentences should be more coherent and logical.
Author Response
Thanks for your comments and suggestions. Please refer to the attached pdf file with responses to the Reviewer

Reviewer 3 Report
Comments and Suggestions for Authors
Comments were addressed.
Comments on the Quality of English LanguageEnglish can be improved.
Author Response

(The authors gave the same response as above.)

Reviewer 4 Report
Comments and Suggestions for Authors
The work was improved, however, I still maintain my opinion that the work would need a greater degree of depth and reasoned analysis about the measurement process and its consequences, presenting an appropriate discussion of the results and evaluating the impacts of the environment. This only presents a process to clean the air, but it will not contribute to a decrease in particle production, and if the particles are not in the air, they will go into the soil. What are the problems with that? Did the authors analyze this? The paper does not present a solution to solve a sustainability problem, this only transfers the problem.
nothing to say.
Author Response

(The authors gave the same response as above.)
